# COVID-19 and Pulmonary Angiogenesis: The Possible Role of Hypoxia and Hyperinflammation in the Overexpression of Proteins Involved in Alveolar Vascular Dysfunction

**DOI:** 10.3390/v15030706

**Published:** 2023-03-08

**Authors:** Anna Flavia Ribeiro Santos Miggiolaro, Felipe Paes Gomes da Silva, David Batista Wiedmer, Thiago Mateus Godoy, Nicolas Henrique Borges, Giulia Werner Piper, Alessandro G. G. Oricil, Carolline Konzen Klein, Elisa Carolina Hlatchuk, Júlio César H. Dagostini, Mariana Collete, Mayara Pezzini Arantes, Raissa C. D’Amico, Anderson A. Dutra, Marina Luise Viola de Azevedo, Lucia de Noronha

**Affiliations:** 1Postgraduate Program of Health Sciences, School of Medicine and Life Sciences, Pontifícia Universidade Católica do Paraná—PUCPR, R. Imaculada Conceição, 1155—Prado Velho, Curitiba 80215-901, PR, Brazil; 2Laboratory of Experimental Pathology, School of Medicine and Life Sciences, Pontifícia Universidade Católica do Paraná—PUCPR, R. Imaculada Conceição, 1155—Prado Velho, Curitiba 80215-901, PR, Brazil; 3Department of Medical Pathology, School of Health Sciences, Universidade Federal do Paraná—UFPR, R. Padre Camargo, 280—Alto da Glória, Curitiba 80060-240, PR, Brazil

**Keywords:** SARS-CoV-2, endothelial dysfunction, angiogenesis, microthrombi, immunohistochemistry

## Abstract

COVID-19 has been considered a vascular disease, and inflammation, intravascular coagulation, and consequent thrombosis may be associated with endothelial dysfunction. These changes, in addition to hypoxia, may be responsible for pathological angiogenesis. This research investigated the impact of COVID-19 on vascular function by analyzing post-mortem lung samples from 24 COVID-19 patients, 10 H1N1pdm09 patients, and 11 controls. We evaluated, through the immunohistochemistry technique, the tissue immunoexpressions of biomarkers involved in endothelial dysfunction, microthrombosis, and angiogenesis (ICAM-1, ANGPT-2, and IL-6, IL-1β, vWF, PAI-1, CTNNB-1, GJA-1, VEGF, VEGFR-1, NF-kB, TNF-α and HIF-1α), along with the histopathological presence of microthrombosis, endothelial activation, and vascular layer hypertrophy. Clinical data from patients were also observed. The results showed that COVID-19 was associated with increased immunoexpression of biomarkers involved in endothelial dysfunction, microthrombosis, and angiogenesis compared to the H1N1 and CONTROL groups. Microthrombosis and vascular layer hypertrophy were found to be more prevalent in COVID-19 patients. This study concluded that immunothrombosis and angiogenesis might play a key role in COVID-19 progression and outcome, particularly in patients who die from the disease.

## 1. Introduction

The COVID-19 (Novel Coronavirus Disease 2019) pandemic began in Wuhan, China, in December 2019 and quickly spread globally. SARS-CoV-2 (Severe Acute Respiratory Syndrome Coronavirus 2) is the virus responsible for the disease and infects through the ACE-2 (Angiotensin-Converting Enzyme-2) receptor, which is expressed in cells, including pneumocytes and endothelial cells. Research is ongoing to improve the diagnosis, prevention, and treatment of COVID-19 [1,2]. Endothelial cells play a crucial role in maintaining blood vessel function and balance in the body. Healthy endothelial cells produce factors that promote blood flow, inhibit blood clotting, and promote fibrinolysis. Dysfunctional endothelial cells lead to a shift towards decreased blood flow and the increased risk of blood clots [3].

In COVID-19, two main pathological effects have been observed in endothelial cells: hypercoagulation and abnormal angiogenesis. The hypercoagulable state is driven mainly by immunothrombosis, triggered by neutrophils and monocytes, leading to the formation of microthrombi in small vessels. Uncontrolled immunothrombosis can result in coagulation cascade dysregulation and microthrombus formation, potentially leading to disseminated thromboinflammation [3,4]. Angiogenesis, the formation of new blood vessels from existing microvascular structures, can be triggered by factors such as inflammation, hypoxia, and oxidative stress. It involves several steps, including degradation of the basement membrane, proliferation, migration, and tube formation of endothelial cells in response to pro-angiogenic factors. The analysis of various immunomarkers is being conducted to study the molecular, tissue, and vascular aspects of COVID-19, with some pointing to endothelial dysfunction, immunothrombosis, and angiogenesis as contributing factors to the development of typical clinical symptoms seen in daily practice [3,4,5].

COVID-19 can range from asymptomatic to ARDS (Acute Respiratory Distress Syndrome), with a high risk of death. Although pneumonia and DAD (Diffuse Alveolar Damage) are the most common symptoms, there is evidence of systemic lesions affecting multiple organs, particularly in non-survivor patients. Among the pathophysiological pathways, vascular involvement is particularly significant. Multiple systems are affected by COVID-19-related vascular dysfunctions, including the pulmonary vasculature, heart, legs, spleen, and brain. These complications often occur in multiple organs and carry a high mortality rate in severe disease cases, with pulmonary embolism being a common thrombotic event in the illness [3,6].

An influenza virus infection can also trigger endothelial cell dysfunction. The influenza A infection of endothelial cells also causes a significant modulation of the Inhibitor of Activation of the Plasminogen-1 (PAI-1). Pronounced changes in the degree of PAI-1 expression in endothelial cells confirm the ability of the Influenza A virus H1N1 subtype (H1N1pdm09) to cause intravascular coagulation [7,8].

The study aims to compare the pulmonary vascular changes caused by COVID-19 with those caused by the H1N1pdm09 and control paraffined lung samples. This will be achieved by analyzing the histopathological and molecular changes in paraffined lung samples using biomarkers/proteins involved in microthrombosis, endothelial dysfunction, and angiogenesis. The aim is to understand the similarities and differences between these two respiratory viruses and their impact on pulmonary vasculature.

## 2. Materials and Methods

### 2.1. Ethical Approvals

The presented study was approved by the National Research Ethics Committee (Conselho Nacional de Ética em Pesquisa—CONEP), protocol numbers 3.944.734/2020 and 2.550.445/2018. The authors confirm that all methods followed relevant guidelines and regulations. The families permitted the post-mortem biopsy of the cases of COVID-19 and H1N1pdm09 and signed the informed consent forms.

### 2.2. Samples

The COVID-19 group consists of post-mortem lung samples from patients whose cause of death was ARDS (*n* = 24) during the COVID-19 pandemic (from April 2020 to August 2020). Clinical data (comorbidities, chronic and hospital use medications, ventilatory parameters, laboratory results, and procedures) were obtained from medical records during hospitalization in the Intensive Care Unit (ICU) at the Hospital Marcelino Champagnat in Curitiba, Brazil.

Testing for COVID-19 was performed on nasopharyngeal swabs taken during ICU hospitalization by Real-Time Polymerase Chain Reaction (RT-qPCR). The viral genome’s amplification was performed with the Invitrogen SuperScript™III Platinum^®^ One-Step qRT-PCR Kit (Catalog number: 11732020, Waltham, MA, USA), and was positive for SARS-CoV-2. 

The H1N1 group consists of post-mortem lung samples from patients whose cause of death was ARDS during the H1N1pdm09 pandemic between July to September 2009 (*n* = 10). The patients were tested through fresh samples of lung post-mortem biopsies, and the performed qRT-PCR (a similar technique to that of the COVID-19 group) was positive for H1N1pdm09.

A minimally invasive lung post-mortem biopsy was performed (in both groups, H1N1 and COVID-19) through an anterior left mini-thoracotomy with upper left lobe lingular segment resection. The resected pieces were 3 × 3 × 2 cm.

The CONTROL group (*n* = 11) comprised necropsy lung samples of patients who died due to cardiovascular and neoplastic disease, not involving lung lesions.

### 2.3. Histopathological and Immunohistochemistry Analysis

All lung samples were formalin-fixed paraffin-embedded (FFPE), taken for microscopic assessment, and underwent routine processing, embedding, sectioning at 4 µm, and staining with hematoxylin and eosin—H&E (Harris Hematoxylin: NewProv, Cod. PA203, Pinhais, BR; Eosin: BIOTEC Reagentes Analíticos, Cod. 4371, Pinhais, BR).

Microthrombosis, endothelial injury and vascular hypertrophy, activators of angiogenesis, were evaluated through histopathological analysis. A pathologist evaluated these histopathological findings using a semiquantitative grading (absence, mild, moderate, severe) in all groups.

The immunohistochemistry technique was applied to identify the immunoexpression of Anti-ICAM-1 (Intercellular Adhesion Molecule-1or CD54), Anti-ANGPT-2 (Angiopoietin-2), Anti-IL-6 (Interleukin-6), Anti-IL-1β (Interleukin-1β), Anti-vWF (von Willebrand Factor), Anti-PAI-1 (Plasminogen Activator Inhibitor-1), Anti-CTNNB-1 (β-Catenin-1), Anti-GJA-1 (Gap Junction Alpha-1 or Connexin-43), Anti-VEGF (Vascular Endothelial Growth Factor), Anti-VEGFR-1 (VEGF Receptor-1), Anti-NF-kB (Nuclear Factor Kappa-Light-Chain-Enhancer of Activated B Cells), Anti-TNF-α (Tumor Necrosis Factor-α), and Anti-HIF-1α (Hypoxia-Inducible Factor-1α), as shown in Appendix A. 

The immunohistochemistry technique recommended an overnight incubation protocol for primary antibodies in a humid chamber between 2 and 8 °C. The secondary polymer (Mouse and Rabbit Specific HRP/DAB IHC Detection Kit—Micro-polymer, Abcam, ab236466, Cambridge, UK) was applied to the material tested for 20 min at room temperature. The technique was revealed by adding the 2, 3, diamino-benzidine complex + hydrogen peroxide substrate, the time required to turn brown, and the counterstaining with Harris Hematoxylin. The reactivity of positive control confirmed the results.

The immunostained slides were scanned with an Axio Scan.Z1 Scanner (ZEISS, Jena, Germany). ZEN Blue Edition (ZEISS, Jena, Germany) was then utilized to randomly generate thirty High-Power Fields (HPF = 40× objective) for the COVID-19 group and twenty HPF for the H1N1 and CONTROL groups. The analysis was blind once the software randomly generated the images, with no investigator interference. The immunopositivity areas were measured by Image-Pro Plus software version 4.5 (Media Cybernetics, Rockville, MD). These areas were subsequently converted into percentages per HPF to enable statistical analysis.

### 2.4. Statistical Analysis

The normality condition was evaluated using the Shapiro-Wilk test. The comparison of quantitative variables between the three groups was obtained using the non-parametric Kruskal-Wallis test. When there was significance, the Mann-Whitney test was performed in the two-by-two comparison. For demographic variables, the Student’s *t*-test was used. Fisher’s exact test was used to study the associations between clinical variables and biomarkers. Values of *p* < 0.05 indicated statistical significance. Data were analyzed using JMP (™) Pro 14.0.0 software.

## 3. Results

The presence of endothelial damage, microthrombosis and vascular hypertrophy, which stimulates angiogenesis, and the histological parameters were evaluated through semiquantitative histopathological analyses (Table 1).

The immunomarkers used to verify endothelial damage, closely related to the angiogenic response, were ICAM-1, ANGPT-2, IL-6, and IL-1β. ICAM-1 showed an increased tissue expression (*p* < 0.0001) in the COVID-19 compared to the CONTROL and H1N1 groups. The ANGPT-2 showed an increased tissue expression (*p* = 0.0055) in the COVID-19 compared to the CONTROL group. The same increased tissue expression of IL-1β was observed in the COVID-19 compared to the CONTROL and H1N1 groups (*p* = 0.0002 and *p* = 0.0221, respectively). These findings are shown in Figure 1.

Regarding the PAI-1, there was a lower tissue expression in COVID-19 compared with the CONTROL group (*p* = 0.0276). These findings are shown in Figure 2.

CTNNB-1 and GJA-1 showed increased tissue expression in the COVID-19 compared to the CONTROL (*p* < 0.0001 and *p* = 0.019, respectively) and H1N1 (*p* < 0.0001 and *p* = 0.005, respectively) groups. When VEGF was analyzed, there was an increased tissue expression in the COVID-19 compared to the CONTROL group (*p* = 0.0024). For the VEGFR-1, there was an increased tissue expression in the COVID-19 compared to the CONTROL group (*p* = 0.0032). Regarding the NF-kB, there was an increased tissue expression in the COVID-19 compared to the CONTROL and H1N1 groups (*p* < 0.0001 and *p* = 0.0002, respectively). Regarding TNF-α, a significantly increased tissue expression was observed in the COVID-19 group compared to the CONTROL group (*p* = 0.0006). These findings are shown in Figure 3.

The clinical data analyzed in this study were comorbidities with probable endotheliopathy (Type 2 Diabetes Mellitus, arterial hypertension, dyslipidemia and obesity), home and hospital use of medications with action on the immune system and coagulation (corticosteroids, anticoagulants and platelet antiaggregant), and ventilatory parameter (PaO2/FiO2 ratio) (Table 2), and they were only compared to the tissue expression of biomarkers of the COVID-19 group.

COVID-19 group patients with Type 2 Diabetes Mellitus (DM2) were associated with a higher expression of vWF (*p* = 0.032) (Appendix A). The study also showed an association between patients with arterial hypertension and dyslipidemia and an increased expression of vWF/CTNNB-1 (*p* = 0.049/*p* = 0.023) and IL-6/GJA-1 (*p* = 0.034/*p* = 0.004), respectively. No associations were found between the presence of obesity and the tissue expression of biomarkers of this study.

The study showed an association between patients who did not receive corticosteroids and an increased expression of IL-6 (*p* = 0.049) within the COVID-19 group (Appendix A). The present study did not find statistical significance when it associated the home use of anticoagulants and antiplatelet agents with the tissue expression of biomarkers, except for IL-6 (*p* = 0.042). In the latter case, the use of antiplatelet agents was associated with higher IL-6 tissue expression (Appendix A).

A significant association was observed between the ventilatory parameter PaO_2_/FiO_2_ < 150 and VEGF expression (*p* = 0.008) within the COVID-19 group (Appendix A). 

## 4. Discussion

### 4.1. Anatomopathological and Immunohistochemical Findings

This work highlights the importance of understanding the mechanisms of COVID-19’s vascular involvement and its effect on multiple organs and systems. The vascular changes related to COVID-19 are not limited to the lungs themselves, but can affect various organs and systems by triggering a procoagulant state, clinically culminating in arterial and venous thrombotic conditions. Vascular damage can be responsible for death and severe consequences for survivors, such as long-term/chronic COVID-19 and the worsening of previous diseases related to endothelial dysfunction [9,10,11,12,13,14,15].

This study found that there are activated endothelial cells and high levels of biomarkers of blood vessel damage (ICAM-1, ANGPT-2, IL-1β) in severe COVID-19 cases. These findings suggest a significant involvement of the blood vessels in the lungs and the possibility of angiogenesis as a result. The increased tissue expression of ICAM-1, ANGPT-2 and IL-1β, and the prominent presence of activated endothelial cells on histopathological examination was observed in comparison to control and H1N1pdm09 patients. The study also found that H1N1pdm09 infection can lead to milder endothelial dysfunction. These findings suggest that COVID-19 can lead to severe vascular compromise, with the presence of microthrombi and the marked expression of inflammatory mediators [7].

There was no significant difference in the tissue expression of IL-6 between the COVID-19, H1N1pdm09, and control patients. This is contrary to previous studies which used a similar method and found a significant difference. However, the results of this study support the idea that both Influenza A and SARS-CoV-2 increase macrophage activation and chemokines expression. IL-6 has been established as a reliable marker for the severity and mortality of COVID-19 in clinical practice, and its antagonist has been shown to reduce mortality in hospitalized COVID-19 patients [5,7,9,16,17,18,19].

Our findings may support the evidence that the NLRP3 inflammasome plays a crucial role in the innate immune response to SARS-CoV-2 virus infection due to the increased tissue expression of IL-1β in the COVID-19 group compared to the CONTROL and H1N1 groups. The NLRP3 inflammasome activation is related to the transcription of the IL-1β gene and the production of Caspase-1, resulting in pyroptosis. The inflammasome activation may indicate a more severe disease and can be a potential therapeutic target. These results are consistent with previous studies, reinforcing the importance of IL-1β in understanding the immune response to COVID-19 [17,20,21,22]. The increased expression of ICAM-1 in the COVID-19 group suggests that the virus causes significant endothelial damage, leading to systemic inflammation and the potential for further vascular remodeling. These findings align with previous studies that have also demonstrated the importance of ICAM-1 as a marker of endothelial damage in COVID-19 patients [4,23]. The high tissue expression of ANGPT-2 in the COVID-19 and H1N1 groups compared to the CONTROL group suggest its involvement in inflammation and vascular damage caused by respiratory infections. ANGPT-2 is being studied as a marker of endothelial injury and severity of COVID-19, and its elevated plasma levels predict worse outcomes. ANGPT-2 may also be associated with necroptosis, and this evidence has driven the therapy research [24,25,26]. The relationship between ANGPT-2-associated endotheliopathy and platelet level changes has been suggested by Price et al. [27]. The findings of this study and previous ones reinforce the need for further studies to better understand the role of ANGPT-2 and other markers in diagnosing and treating respiratory infections such as COVID-19 and H1N1pdm09 [24,25,26,27,28].

Our work found that a large proportion of COVID-19 patients (83.33%) have microthrombosis in their lungs, compared to much lower levels in the CONTROL (9%) and H1N1 (10%) groups. The study suggests that SARS-CoV-2 can lead to uncontrolled thrombogenic activity and pose a threat to life, whereas other infections such as Influenza A can also cause blood vessel dysfunction [7,8]. The final formation of microthrombi has multiple complex pathways that are still being studied and elucidated, but COVID-19 is already recognized as a disease that is characterized by hypercoagulability, inflammation, and vascular dysfunction. Other studies have mentioned microthrombosis in post-mortem lung samples from COVID-19 carriers [15,29,30,31]. This result suggests that COVID-19 is connected to a pathological immune response leading to microthrombosis in the lungs’ blood vessels. This process is triggered by activating the body’s innate immune system in response to the virus in the bloodstream, releasing substances that promote thrombosis and inhibit anticoagulant pathways. Other factors include the activation of a protein called vWF, the complement system, and platelets. While this immune response is natural, it can become harmful when it occurs excessively. Severe COVID-19 cases often show these uncontrolled immune-thrombogenic reactions, including hyperactivation of the innate immune system, endothelial dysfunction, activation of the inflammasome system, reduction of fibrinolysis, platelet activation, hypoxia, and the presence of neutrophil extracellular traps (NETs) [4,32].

The histological finding of microthrombosis led this study to evaluate the tissue expression of vWF and PAI-1. Despite being widely expressed in COVID-19, the study found no significant difference in vWF expression compared to samples from patients with H1N1pdm09 and control. vWF plays a role in platelet adhesion during blood vessel injury. Systemic inflammation can trigger its rapid and intense release, considerably increasing plasma levels and reducing tissue deposits. This observation may support our results, reinforced by the fact that severe COVID-19 usually presents a vWF significant increase in plasma levels, with apparently reduced tissue distribution. Elevated vWF levels have been linked to endothelial injury, disease severity, and increased mortality in COVID-19 patients [23,29,33,34,35,36]. PAI-1 plays a crucial role in COVID-19, contributing to the prothrombotic state characteristic of the disease. PAI-1 expression is upregulated in pro-inflammatory environments, which is seen in COVID-19 as well. PAI-1 blocks PIAS3 (activated STAT-3 protein inhibitor) and induces an increase in the secretion of pro-inflammatory cytokines, perpetuating the prothrombotic state. Current guidelines for the prevention and treatment of COVID-19 are limited to heparins and mechanical methods. However, discovering new therapeutic modalities targeting the STAT (Signal Transducer and Transcription Activator) system can enhance these guidelines and help reduce thrombus formation [3,37,38,39]. Few studies address the expression of PAI-1 in tissues carrying H1N1pdm09. However, a study noted that cells infected by this virus showed a PAI-1/tPA interaction, concluding that this pathway would participate in the pathophysiological mechanisms of coagulopathy in severe cases [8].

All biomarkers discussed above (ICAM-1, ANGPT-2, IL-1β, IL-6, vWF and PAI-1) participate in pathological processes that stimulate responses such as angiogenesis. The histopathological results of this study showed that COVID-19 is associated with the hypertrophy of the vascular layers, with 91.66% of COVID-19 samples showing this change compared to only 20% in the H1N1 and 0% in the CONTROL group. This increase in vascular layer thickness may result from hypoxia and inflammation or a reaction to intense endothelial injury that causes dysfunction in the underlying vascular wall. These findings highlight the importance of further research into the mechanisms behind these changes and how they may contribute to the development of new treatments for COVID-19 [40].

Our work also observed a significant increase in the tissue expression of TNF-α and NF-kB in the COVID-19 group compared to the CONTROL group. This suggests that the TNF-α/TNFR1/NF-kB mechanism, which involves the activation of M1 macrophages and the release of pro-inflammatory agents, may play a role in angiogenesis in patients infected with SARS-CoV-2. This mechanism may also be activated in response to hypoxic conditions and promote the transcription of genes involved in regulating the vascular endothelium, such as HIF-1α [3,41,42,43,44].

Both tissue expression of VEGF and VEGFR-1 was significantly higher in the COVID-19 than in the CONTROL group, and tissue expression of VEGF was also significantly higher compared to the H1N1 group. The findings suggest that the activation of the VEGF/VEGFR-1 pathway may play an essential role in the angiogenic process in COVID-19 patients, contributing to the development of new vessels in the areas of hypoxia, inflammation and viral replication, which in turn contributes to the increase in vascular hypertrophy observed in this study. The trigger for its overexpression is the response to episodes of hypoxia and pro-inflammatory states caused by cytokine storms. It is responsible for migrating new endothelial cells and inhibiting cell death, favoring angiogenesis [45,46,47]. Studies also demonstrate the secretion of HIF-1α in response to the hypoxemic environment seen during SARS-CoV-2 infection, in addition to the pro-inflammatory state. Responding to this environment, the endothelial cells are activated and bind to VEGF through VEGFR-1, contributing to signaling positive for angiogenic events. VEGF upregulation is also related to COVID-19 progression factors, where elevated levels of the VEGF-A Flt-1/VEGFR-1 complex are correlated with disease severity [5,15,48,49].

We also evaluated the immunoexpression of GJA-1 (Connexin-43). This protein is characteristic of a gap junction of endothelial cells that allows them to communicate along the vascular wall. It is present and expressed in angiogenic environments and is related to the progression of vascular diseases. Our results suggest that the increase in tissue expression of GJA-1 in the COVID-19 group may play a role in forming new vessels, contributing to the angiogenic response in the hypoxemic and pro-inflammatory environment. The results align with previous studies, which have shown the higher expression of GJA-1 (Connexin-43) in SARS-CoV-2 infected cell cultures [50].

There was greater tissue expression of CTNNB-1 (β-catenin) in COVID-19 samples compared to the CONTROL and H1N1 groups. Studies have demonstrated the role of CTNNB-2 (β-catenin) in inflammation and repair promoted by alveolar macrophages during COVID-19, finding a relationship between this protein and HIF-1α in the sense of intensifying inflammatory activity [51].

The subpopulation of samples in this study was also used in a paper by Kulasinghe et al. [52] in 2021. They analyzed mRNA from over 1800 genes using NanoString GeoMX DSP and bioinformatic modeling, and selected samples with the greatest severity. The Kulasinghe et al. study used 10 COVID-19, five H1N1pdm09, and four control patients. Four proteins (CTNNB1, HIF-1α, VEGF and ANGPT) related to the 22 mRNAs described for COVID-19 were analyzed. They showed increased gene expression in the COVID-19 group, consistent with the idea that SARS-CoV-2 infection-caused hypoxia stimulates the secretion of angiogenesis-related proteins.

Our study found significant differences in the tissue expression of various biomarkers involved in angiogenesis between the COVID-19, H1N1, and CONTROL groups. The increased expression of TNF-α, NF-kB, VEGF, VEGFR-1, GJA-1 (Connexin-43), and CTNNB-1 (β-catenin) in the COVID-19 group suggests a strong activation of angiogenic pathways in response to hypoxia and hyperinflammation associated with SARS-CoV-2 infection. The comparison of the results from the COVID-19 group with those of the H1N1 group suggests that hypoxia may not be the main trigger of endothelial dysfunction and angiogenesis. This is because both groups underwent mechanical ventilation, and the expression of HIF-1 was similar between them. As a result, hyperinflammation and the presence of SARS-CoV-2 may play a key role. These findings contribute to a better understanding of the mechanisms involved in angiogenesis in the context of COVID-19.

### 4.2. Clinical Findings

Our clinical findings suggest that COVID-19 increases the risk of thromboembolic events in patients with DM2, mainly through inflammatory pathways (immunothrombosis). DM patients already have a chronic inflammatory state, which is exacerbated in the presence of COVID-19, leading to increased thrombotic risk. The study found a significant association between DM2 and vWF, but no significant association with VEGF/VEGFR-1 markers. Endothelial damage and immunothrombosis, which are part of vascular alterations, can be precursors of angiogenesis [53]. 

This study found no significant association between the home use of antiplatelet agents and the tissue expression of biomarkers except for IL-6. This may indicate that these drugs do not provide extra protection against thromboembolic events. Studies on patients who use oral anticoagulants chronically have also not shown a protective effect on mortality or thrombotic complications in hospitalized patients [54,55,56]. This work found no significant association between the use of anticoagulants and biomarkers, reinforcing the questionable benefit of these drugs in reducing mortality or preventing venous thromboembolism [57,58,59,60].

The administration of corticosteroids during COVID-19 patients’ hospitalization and its association with tissue immunomarker expression was analyzed. The study found an association between patients who did not receive corticosteroids and the increased expression of IL-6. Studies have shown the benefits of corticosteroids, such as dexamethasone, in reducing mortality in COVID-19 patients and reducing biomarkers such as ICAM-1 and ANGPT-2 [61,62,63,64]. Our results suggest that corticosteroids may benefit not only in reducing inflammation but also in inhibiting endothelial dysfunction and angiogenesis.

Our study identified a significant association between the ventilatory parameter of PaO2/FiO2 ratio <150 (indicative of moderate to severe hypoxia at the time of orotracheal intubation [65]) and VEGF expression. This marker characterizes the potential angiogenic present in COVID-19 samples and its relationship with hypoxia, identified through bedside ventilatory parameters. Evidence has shown that COVID-19 patients are at increased risk of thrombotic events when they have a lower PaO_2_/FiO_2_ ratio on admission [66,67]. This highlights the importance of monitoring oxygenation levels in COVID-19 and the potential benefit of using drugs targeting angiogenesis and thrombotic events in these patients. The relationship between hypoxia and VEGF expression may also have implications for developing new treatments to reduce angiogenesis in COVID-19 patients. 

### 4.3. Limitations of the Study

This study has limitations such as a limited sample size, the fact that the tissue changes are only seen at the time of death and may be influenced by other factors, and differences in age and length of hospital stay, among the groups. The control patients died from diseases other than pulmonary infection, but these differences may still have impacted the results. However, despite these limitations, the study provides evidence that COVID-19 is associated with significant vascular changes compared to other viral respiratory infections and death from other causes.

## 5. Conclusions

The findings of our study suggest that severe COVID-19 patients experience alveolar vascular dysfunctions (thrombosis, endothelial injury, alteration of vascular wall integrity, angiogenesis), which may impact other organs and systems in the body. These dysfunctions are a result of a complex interplay between viral tissue action, hypoxia, and hyperinflammation. Drugs that target biomarkers such as the ICAM-1, ANGPT-2, IL-1β, PAI-1, VEGF, VEGFR-1, GJA-1, CTNNB-1, and NF-kB pathways may prove to be effective in preventing and treating the consequences of these dysfunctions. Finally, clinical data, including anticoagulants or antiplatelet drugs and bedside ventilatory and hypoxia parameters, can inform treatment decisions and help prevent complications related to vascular injuries.

## Figures and Tables

**Figure 1 viruses-15-00706-f001:**
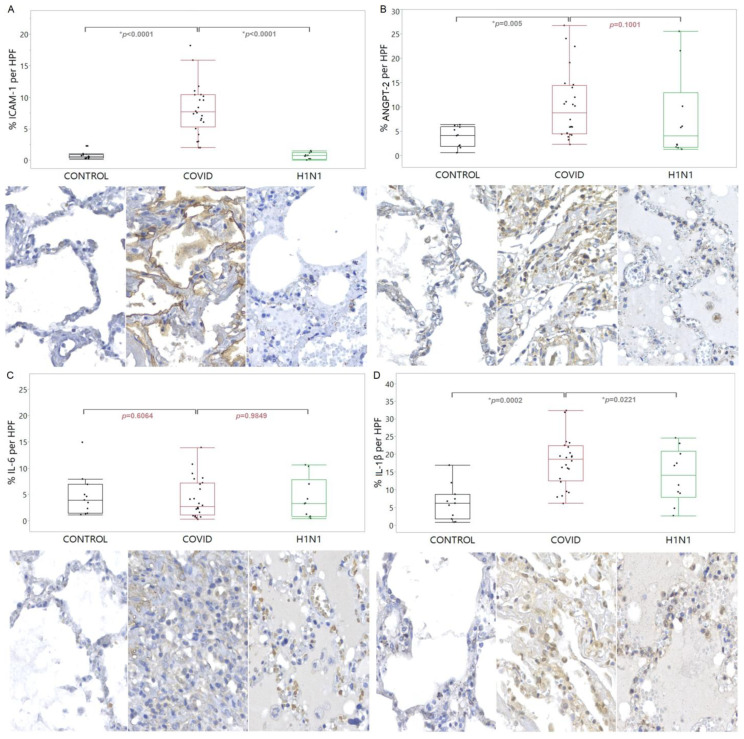
Compares the levels of ICAM-1 (**A**), ANGPT-2 (**B**), IL-6 (**C**), and IL-1β (**D**) proteins in tissue samples from COVID-19 patients versus control and H1N1pdm09 patients. The data is presented as a graphical representation of the comparison between the groups and the corresponding immunohistochemistry staining images. The difference between the groups was determined using the non-parametric Kruskal-Wallis test. When there was significance, the Mann-Whitney test was applied in the two-by-two comparison, with significance (*) indicated by a *p*-value less than 0.05.

**Figure 2 viruses-15-00706-f002:**
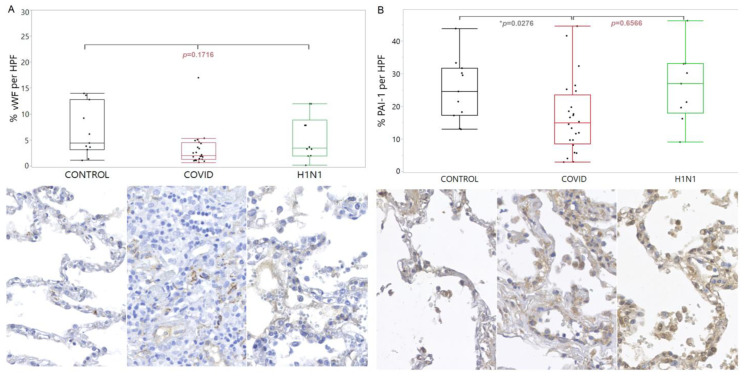
Comparison of the levels of vWF (**A**) and PAI-1 (**B**) proteins in tissue samples from COVID-19 patients versus control and H1N1pdm09 patients. The data is presented as a graphical representation of the group comparison and the corresponding immunohistochemistry staining images. The difference between the three groups was determined using the non-parametric Kruskal-Wallis test. When the three groups were significant, the Mann-Whitney test was applied in the two-by-two comparison, with significance (*) indicated by a *p*-value less than 0.05.

**Figure 3 viruses-15-00706-f003:**
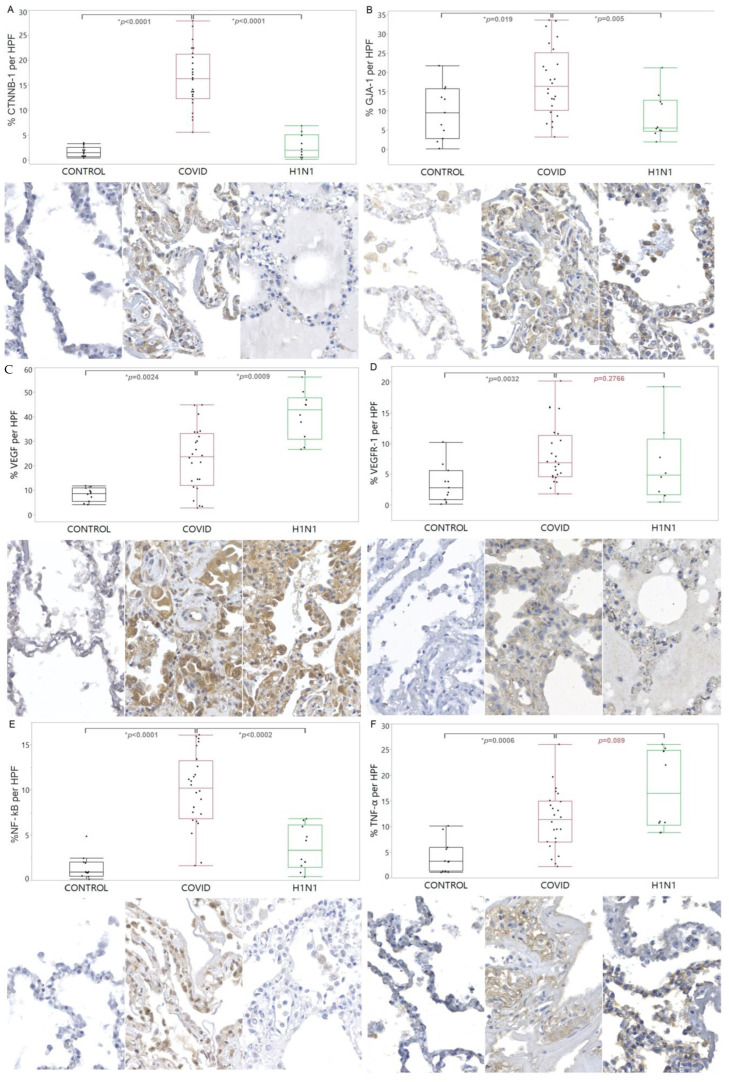
Comparison of the levels of CTNNB-1 (**A**), GJA-1 (**B**), VEGF (**C**), VEGFR-1 (**D**), NF-kB (**E**), and TNF-α (**F**) proteins in tissue samples from COVID-19 patients versus control and H1N1pdm09 patients. The data is presented as a graphical representation of the comparison between the groups and the corresponding immunohistochemistry staining images. The difference between the groups was determined using the non-parametric Kruskal-Wallis test. When there was significance, the Mann-Whitney test was applied in the two-by-two comparison, with significance (*) indicated by a *p*-value less than 0.05.

**Table 1 viruses-15-00706-t001:** Comparative Table between COVID-19, H1N1 and CONTROL groups for gender, age, time from hospitalization to death, mechanical ventilation time, and histological patterns of DAD.

Data	Variable		CONTROL (*n* = 11)	COVID-19 (*n* = 24)	H1N1 (*n* = 10)
Gender ^1^	Male		72.7% (*n* = 8)	62.5% (*n* = 15)	80% (*n* = 8)
	Female ** p*-values		27.3% (*n* = 3) *p* = 0.709	37.5% (*n* = 09)	20% (*n* = 2) *p =* 0.437
Age (years) ^2^	----- ** p*-values		42.3/45 (18–60) *p* < 0.001	71.96/72.5 (46–93)	43.5/44 (23–61) *p* < 0.001
Time from hospitalization to death (days) ^2^	----- ** p*-values		7.6/4 (1–46) *p* < 0.003	15.87/13 (1–39)	4.70/1.5 (1–19) *p* < 0.003
Mechanical ventilation (days) ^2^	----- ** p*-values		----- -----	12.04/9.5 (0–36)	4.70/1.5 (1–19) *p =* 0.028
Histological patterns of DAD	-----		Normal septum	Interstitial pneumonitis with scarce septal neutrophils with hyaline membrane and micro thrombosis	Interstitial pneumonitis with high septal neutrophils infiltration and no micro thrombosis
Semiquantitative Grading ^1^	Micro thrombosis	absence	91% (*n* = 10)	16.67% (*n* = 4)	90% (*n* = 9)
		mild	9% (*n* = 1)	37.5% (*n* = 9)	10% (*n* = 1)
		moderate	0% (*n* = 0)	33.33% (*n* = 8)	0% (*n* = 0)
		severe	0% (*n* = 0)	12.5% (*n* = 3)	0% (*n* = 0)
		** p*-values	*p* < 0.0001		*p* < 0.0001
	Endothelial Damage	absence	45.45% (*n* = 5)	0% (*n* = 0)	0% (*n* = 0)
		mild	54.55% (*n* = 6)	4.17% (*n* = 1)	60% (*n* = 6)
		moderate	0% (*n* = 0)	20.83% (*n* = 5)	40% (*n* = 4)
		severe	0% (*n* = 0)	75% (*n* = 18)	0% (*n* = 0)
		** p*-values	*p* < 0.0001		*p* < 0.0001
	Vascular Hypertrophy	absence	100% (*n* = 11)	8.34% (*n* = 2)	80% (*n* = 8)
		mild	0% (*n* = 0)	33.33% (*n* = 8)	10% (*n* = 1)
		moderate	0% (*n* = 0)	33.33% (*n* = 8)	10% (*n* = 1)
		severe	0% (*n* = 0)	25% (*n* = 6)	0% (*n* = 0)
		* *p*-values	*p* < 0.0001		*p* < 0.0001

Legend: ^1^ %(number of cases); ^2^ Average/Median (Min-Max); DAD: diffuse alveolar damage; * = *p*-values for gender, age, time from hospitalization to death, mechanical ventilation and semiquantitative grading were obtained between COVID-19 vs. CONTROL and COVID-19 vs. H1N1 group. *p*-values were performed using the non-parametric Kruskal-Wallis test.

**Table 2 viruses-15-00706-t002:** Clinical data of the COVID-19 group.

COMORBIDITIES	
Diabetes Mellitus (Type 2) ^1^	11 (45.8 %) ^2^
Systemic Arterial Hypertension ^1^	21 (87.5%) ^2^
Dyslipidemia ^1^	17 (70.8%) ^2^
Obesity ^1^	6 (25%) ^2^
Chronic Heart Disease ^1^	11 (45.8%) ^2^
Chronic Lung Disease ^1^	5 (20.8%) ^2^
Malignancy ^1^	3 (12.5%) ^2^
Time from symptom onset to death (days) ^3^	18.6 (07–42) Median = 17
CHRONIC USE MEDICATIONS	
Anticoagulant (use chronic) ^1^	4 (16.6%) ^2^
Antiaggregant Platelets (use chronic) ^1^	15 (62.5%) ^2^
MECHANICAL VENTILATION DATA	
Time of Mechanical Ventilation (MV) ^4^	12.4 (0–29) Median = 11.5
PaO_2_/FiO_2_ Ratio (1ºday MV) ^4^	159.9 (0–313) Median = 127.5
LABORATORIES DATA	
C-Reactive Protein (mg/dL) ^5^	172.35 (8.7–420) Median = 154.2
D-dimer (µg/L) ^5^	14,784 (425–152,174) Median = 1848
Blood platelets (/mm³) ^5^	197,600 (38,000–366,000) Median = 195,000
MEDICATIONS ADMINISTERED DURING HOSPITAL STAY	
Anticoagulant ^1^	24 (100%) ^2^
Corticosteroids ^1^	17 (70.8%) ^2^
Oseltamivir ^1^	08 (33.3%) ^2^
Antibiotic ^1^	24 (100%) ^2^
Hydroxychloroquine and azithromycin ^1^	5 (20.8%) ^2^

Legend: ^1^ Absolute value of N (sample number); ^2^ Percentage; ^3^ Time from symptom onset to death are expressed in days; ^4^ Mechanical ventilation data are expressed post-hospitalization days; ^5^ Laboratory data are expressed according to the units described in the Table.

## Data Availability

The datasets have been attached in Appendix A and are available from the corresponding author upon reasonable request.

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
