# Peer review of "COVID-19 and Pulmonary Angiogenesis: The Possible Role of Hypoxia and Hyperinflammation in the Overexpression of Proteins Involved in Alveolar Vascular Dysfunction"

_viruses, 2023, doi:10.3390/v15030706_

Round 1
Reviewer 1 Report
The proposed study provides new insight into hypoxia induced pulmonary endothelial and vascularization dysfunctions induced by severe Covid-19 infection. Overall a nice study, I just have two comments.
1) A thorough English check-up is essential (preferably by a native speaker). Especially in the results section there are sentences hard to understand, and also redundancies within the paragraphs during data presentation. The whole results section needs to be rephrased entirely.
2) Presenting local pulmonary inflammatory molecule protein expression would even more intersting to be compared with serum cytokine levels. If such data exist, I highly recommend to add them to this study in a separate table.
Author Response
Point 1: A thorough English check-up is essential (preferably by a native speaker). Especially in the results section there are sentences hard to understand, and also redundancies within the paragraphs during data presentation. The whole results section needs to be rephrased entirely.
Response 1: We appreciate your note and provide the English revision
Point 2: Presenting local pulmonary inflammatory molecule protein expression would even more intersting to be compared with serum cytokine levels. If such data exist, I highly recommend to add them to this study in a separate table.
Response 2: Lung tissue samples from patients were collected post-mortem. Unnafortunaly, we do not have the serum cytokine levels.

Reviewer 2 Report
This is an important study that provides information about the characteristics of the lungs of patients who died of COVID-19.
Please change the title as this study did not elucidate the role of anything with causal relationships.
Line 82: Please rephrase “Ethical approves”
It is unclear for which groups “p=0.1716” is indicated.
In some figures, it may be helpful to provide statistical analysis results between control and H1N1.
Where is the HIF-1 data?
Author Response
Point 1: Please change the title as this study did not elucidate the role of anything with causal relationships.
Response 1: We appreciate your observation but understand that the title denotes that hypoxia and hyperinflammation, caused by the viral infection, may be a trigger for the overexpression of inflammatory and angiogenic proteins, and this is demonstrated in the statistical difference between the studied groups. However, we have changed the title to describe this hypothesis better.
Point 2: Line 82: Please rephrase "Ethical approves.”
Response 2: Thank you for your input, and we will adapt the paragraph. "The authors confirm that all methods followed relevant guidelines and regulations. The families permitted the post-mortem biopsy of the cases of COVID-19 and H1N1pdm09 and signed the informed consent forms."
Point 3: It is unclear for which groups "p=0.1716" is indicated.
Response 3: This value (p=0.1716) refers to the result of the non-parametric Kruskal-Wallis test. This test compares the three work groups. If this test shows a statistical difference, we compare two by two using the Mann-Whitney test. In the case of the von Willebrand Factor, the Kruskal-Wallis test showed no difference, and therefore the next step was not performed. We have rewritten the legend of figure 2 to describe this method.
Point 4: In some figures, it may be helpful to provide statistical analysis results between control and H1N1.
Response 4: We understand your point and thank you for your placement, but this data was not added because the H1N1 group, in this work, is acting as positive control while the CONTROL group is playing a negative role. The objective was to compare a group of samples without viral infection and another with a viral infection, also pandemic but not COVID-19, to understand the difference in the inflammatory trigger attributed to Sars-CoV-2.
Point 5: Where is the HIF-1 data?
Response 5: In the last paragraph of the discussion, in sub-item 4.1, we commented that hypoxia might not be the main trigger for endothelial dysfunction and angiogenesis since HIF-1α values did not show a statistical difference when comparing viral groups (COVID versus H1N1). In this case, we chose not to present the HIF-1α data since there was no statistical significance. However, this data is in Supplemental Table 2
